# Post-Traumatic Stress Disorder and Associated Factors during the Early Stage of the COVID-19 Pandemic in Norway

**DOI:** 10.3390/ijerph17249210

**Published:** 2020-12-09

**Authors:** Tore Bonsaksen, Trond Heir, Inger Schou-Bredal, Øivind Ekeberg, Laila Skogstad, Tine K. Grimholt

**Affiliations:** 1Department of Health and Nursing Sciences, Faculty of Social and Health Sciences, Inland Norway University of Applied Sciences, 2418 Elverum, Norway; 2Faculty of Health Studies, VID Specialized University, 4306 Sandnes, Norway; 3Norwegian Center for Violence and Traumatic Stress Studies, 0484 Oslo, Norway; trond.heir@medisin.uio.no; 4Institute of Clinical Medicine, University of Oslo, 0450 Oslo, Norway; 5Faculty of Medicine, University of Oslo, 0372 Oslo, Norway; i.s.bredal@medisin.uio.no; 6Division of Mental Health and Addiction, Oslo University Hospital, 0424 Oslo, Norway; oeekeber@online.no; 7Department of Research, Sunnaas Rehabilitation Hospital HF, 1453 Bjørnemyr, Norway; uxlask@sunnaas.no; 8Department of Nursing and Health Promotion, Faculty of Health Sciences, Oslo Metropolitan University, 0167 Oslo, Norway; 9Faculty of Health Studies, VID Specialized University, 0370 Oslo, Norway; tine.grimholt@vid.no; 10Department of Acute Medicine, Oslo University Hospital, 0424 Oslo, Norway

**Keywords:** coronavirus, mental health, Norway, population study, PTSD

## Abstract

The COVID-19 outbreak and the sudden lockdown of society in March 2020 had a large impact on people’s daily life and gave rise to concerns for the mental health in the general population. The aim of the study was to examine post-traumatic stress reactions related to the COVID-19 pandemic, the prevalence of symptom-defined post-traumatic stress disorder (PTSD), and factors associated with post-traumatic stress in the Norwegian population during the early stages of the COVID-19 outbreak. A survey was administered via social media channels, to which a sample of 4527 adults (≥18 years) responded. Symptom-defined PTSD was measured with the PTSD Checklist for the DSM-5. The items were specifically linked to the COVID-19 pandemic. We used the DSM-5 diagnostic guidelines to categorize participants as fulfilling the PTSD symptom criteria or not. Associations with PTSD were examined with single and multiple logistic regression analyses. The prevalence of symptom-defined PTSD was 12.5% for men and 19.5% for women. PTSD was associated with lower age, female gender, lack of social support, and a range of pandemic-related variables such as economic concerns, expecting economic loss, having been in quarantine or isolation, being at high risk for complications from COVID-19 infection, and having concern for family and close friends. In conclusion, post-traumatic stress reactions appear to be common in the Norwegian population in the early stages of the COVID-19 outbreak. Concerns about finances, health, and family and friends seem to matter.

## 1. Introduction

As the COVID-19 outbreak reached Norway in late February 2020 [1], severe suffering and a high number of deaths due to the coronavirus in countries such as China and Italy had already been reported. To curb the spread of the virus, Norwegian authorities imposed a lockdown of society on 12 March 2020. While restrictions were lifted gradually during the following months, “social distancing” became the main policy for public behavior [2,3]. In general, people were encouraged to shelter in place. Kindergartens, schools, and universities were closed, as were non-vital businesses requiring physical proximity, and cultural events and travels were cancelled. Consequently, many businesses experienced financial problems, and many employees were furloughed [4]. As a result, both the pandemic itself, as well as the public policies implemented to address it, have given rise to concerns about the public mental health [5,6,7,8,9].

Post-traumatic stress disorder (PTSD) may develop after exposure to exceptionally threatening or horrifying events [10]. Several recent studies have investigated PTSD in the specific context of the coronavirus outbreak. Among patients recovering from the coronavirus disease, the proportion with symptom-defined PTSD have varied between 12.4% and 31.0% [11,12,13,14,15]. Liu and co-workers [11] found that the risk of PTSD was associated with disease severity and perceived stigma related to the disease, whereas Cai and co-workers [12] found lower PTSD rates among patients over 60 years old. In Norway, a recent nationwide study of healthcare and public service providers showed a prevalence of current PTSD for 11.7% of the sample, and that PTSD symptom levels were higher for those working directly with COVID-19 infected patients [16]. Overall, the results suggest that patients with COVID-19 infection, as well as frontline personnel, have increased risk of PTSD and that interventions to support their mental health are appropriate.

The high rates of severe illness and death surrounding the outbreak of COVID-19, and the subsequent lockdown and social distancing policies in Norway, indicate that the pandemic outbreak may be perceived as a traumatic event. As a result, PTSD in the population may be frequent during the pandemic and may be linked to specific concerns related to the COVID-19 context. While several studies have examined PTSD in the coronavirus context [11,12,13,14,15,16], many are hampered by relatively small sample sizes and samples representing specific contexts such as hospitals or geographical regions. Moreover, the research focus has largely been placed on infected patients. Studies of COVID-19-related PTSD in the general population are beginning to emerge. The prevalence of PTSD in general population samples has been found to vary widely (7–54%) across studies and contexts [17], and a rapid review and meta-analysis found the pooled prevalence to be 24% [18]. Thus, it appears that the potentially traumatizing impact of the pandemic extends to the general population and not only to those who are most directly affected. However, we have not been able to locate studies of PTSD in the Norwegian general population during the COVID-19 situation. Knowledge of the mental health of the population and what affects it can have implications for planning measures against the pandemic and its harmful effects.

We postulated that people’s concerns about their own health, the health of family and friends, as well as finances and financial losses were related to post-traumatic stress. Based on results from previous studies in the Norwegian population [19], we considered possible confounders such as age, gender, education, and life situation.

### Aim of the Study

The aim of this study was to examine the prevalence of current symptom-defined PTSD and associated factors in the Norwegian population during the early stages of the COVID-19 outbreak.

## 2. Materials and Methods

### 2.1. Design

The Norwegian cross-sectional survey CORONAPOP collected data by means of an open web-link between 8 April 2020 and 20 May 2020. The web-link was disseminated from several institutions, including The Oslo University Hospital, Sunnaas Hospital, and University of Oslo. In addition, Facebook, Twitter, LinkedIn, and Instagram were used to disseminate links to the survey. The research project, including a link to the survey, was also presented in national and local newspapers.

### 2.2. Sample

The study participants were Norwegian citizens aged 18 years or older. There were no exclusion criteria.

### 2.3. Measures

Sociodemographic and health-related data were collected as self-report measures. The survey employed several measures identical to the ones used in the Norwegian Population (NORPOP) health survey, which was conducted as a postal survey in 2014–2015 [19,20,21,22]. 

#### 2.3.1. Sociodemographic Variables

Data were collected for age groups (18–29 years, 30–39 years, 40–49 years, 50–59 years, 60–69 years, and 70 years or older), gender (male/female), highest completed educational level (high school or lower versus higher education), current employment status (working/paid leave/in education versus not), cohabitation status (living with spouse or partner versus not), and size of place of residence (<200 inhabitants, 200–19,999 inhabitants, 20,000–99,999 inhabitants, 100,000 inhabitants or more). Social support was assessed with the following question: “Do you have friends that will provide help when you need it?” Response options were yes or no.

#### 2.3.2. Problems Related to COVID-19

The participants were asked to indicate (yes or no) whether they in relation to COVID-19 had economic concerns, expected economic loss, had been infected with the coronavirus, had been quarantined or in isolation, self-identified themselves as being at risk of complications if they were to become infected by the coronavirus, and whether they had concerns for family and friends.

#### 2.3.3. The PTSD Checklist for DSM-5 (PCL-5)

For the present study, data were collected on current post-traumatic stress symptoms associated with the COVID-19 outbreak. To achieve a possible PTSD diagnosis, a respondent had to fulfill the DSM-5 symptom criteria for PTSD, except from the A criterion (having experienced accidental or violent death, threat of life, serious injury, or sexual violence) [10]. In the absence of clinical interviews, we used survey data on the PTSD Checklist (PCL-5) to measure symptoms. The 20-item PCL-5 is a self-administered questionnaire assessing the full domain of the DSM-5 PTSD diagnosis [23]. The instrument has four subscales, corresponding to each of the symptom clusters in the DSM-5. The symptoms endorsed were specifically linked to the COVID-19. Each item was scored on a 5-point scale (0 not at all, 1 a little, 2 moderately, 3 quite a bit, 4 extremely) to rate the extent to which the relevant symptom had bothered the person during the past month.

DSM-5 diagnostic guidelines [10] were applied to the PCL-5 to categorize participants as fulfilling PTSD symptom criteria or not. Participants indicating a score of 2 or above on at least one of five re-experiencing symptoms, one of two avoidance symptoms, two of seven symptoms of negative alterations in cognition and mood, and two of six arousal symptoms were classified as fulfilling the PTSD symptom criteria [23,24]. The Norwegian version of the PCL-5 was developed through an alternating procedure of translations and back-translations [25]. The original authors approved the final English back-translation. PCL-5 has good or excellent internal consistency, reliability, and validity [23,24,26].

### 2.4. Statistical Analyses

Prevalence rates of symptom-defined PTSD are presented for men and women separately as numbers and percentages. Cases with positive scores on the critical number of items in each symptom cluster were considered symptom-defined PTSD. Those that could not have reached the critical number of positive items regardless of scores on missing items were considered non-PTSD. We used single and multiple logistic regression analyses to examine the associations between sociodemographic variables and variables related to COVID-19 and PTSD. Multiple logistic regression analysis was used to adjust for age, sex, and social support. All tests were two-tailed, and differences were considered significant if *p* < 0.05. Data were analyzed using SPSS version 26 for Windows [27].

### 2.5. Ethics

The questionnaires were completed anonymously. Ethical approval for conducting the study was granted from the Regional Committee for Medical and Healthcare Ethics (REK no. 130447). The principles in the Declaration of Helsinki were respected.

## 3. Results

### 3.1. Responders

Altogether, 4527 individuals participated in the study. Table 1 shows the sociodemographic characteristics, variables related to COVID-19, and rates of symptom-defined PTSD in the whole sample and by gender. The respondents included a majority of women (85%) and persons with higher education (76%). A small minority (1.4%) had been infected with COVID-19, whereas almost a quarter (23.4%) self-identified themselves as being at risk of complications if they were to become infected with the virus.

Symptom-defined PTSD was more prevalent among women (19.5%) compared to men (12.5%, *p* < 0.001). The recruitment of men and women gave different distributions of age, education, and living conditions (Table 1). More women than men reported that they had friends who would provide them support if needed and women were more likely to be concerned about family and close ones. Conversely, men were more likely to expect economic loss due to the pandemic, and more men than women reported to be at risk of complications if they were to become infected with the coronavirus.

### 3.2. Factors Associated with PTSD

Table 2 shows the associations between the independent variables and symptom-defined PTSD. The risk of PTSD was higher for those of younger age, women, and those without supporting friends. When running the logistic regression model with all predictors included, several problems related to COVID-19 were found to be associated with higher risk of PTSD, such as having economic concerns (OR: 2.23, *p* < 0.001), expecting economic loss (OR: 1.39, *p* < 0.01), having been in quarantine or isolation (OR: 1.31, *p* < 0.01), self-identifying themselves as having high risk of complications if they were to become infected with the coronavirus (OR: 1.77, *p* < 0.001), and being concerned about family and close friends (OR: 1.83, *p* < 0.001).

## 4. Discussion

In this Norwegian population study conducted during the early stages of the COVID-19 outbreak, post-traumatic stress reactions fulfilling the PTSD symptom criteria were found in 19.5% of women and 12.5% of men. Fulfilling the PTSD symptom criteria was associated with lower age, female gender, lack of social support, and a range of problems directly related to the coronavirus outbreak, such as economic concerns and concern about family and friends. 

Thus, a substantial proportion of people experiencing the COVID-19 outbreak in Norway had developed stress reactions that met the symptom criteria for PTSD. Still, the prevalence rates for men and women were lower than the 24% pooled estimate reported across several studies from other countries [18]. However, studies have found widely differing prevalence rates of PTSD between countries, ranging between 7% and 53% [17]. Several factors can explain the large differences in reported stress reactions between populations. These include the numbers of infected and dead, accessibility and capacity in the healthcare system, access to factual or intimidating information, opportunities for financial assistance, the authorities’ handling of the epidemic, and people’s trust in the authorities, as well as methodological differences between studies and regional differences in reporting patterns [17]. Compared with many other countries at the same time, Norway had low infection rates, low or no excess mortality from the pandemic, sufficient capacity in a free healthcare system, state financial support for many needy, and generally high confidence in the authorities.

The finding that younger persons were more inclined to have symptoms surpassing the PTSD threshold is in line with some [12] but not all [28] studies from China. The finding is intriguing, considering the objectively higher risk of complications among older people. While it is possible that younger persons were more inclined to have concerns about family or to fear economic hardship, the association between younger age and PTSD was retained when these factors were controlled for. Thus, in some contexts, older age may serve as an independent resource for coping with adversity in life, as illustrated by studies of the Norwegian general population demonstrating less depression [29] and anxiety [30] among people of older age.

While about one in four expected the pandemic to cause economic loss, about one in five reported that the pandemic gave rise to concerns about their economic situation. Despite substantial overlap (70.3%), we noted that economic concerns appear to induce more stress reactions than the mere prospects of facing economic loss. Wealthy people may face economic loss without major worries, whereas the less affluent may foresee serious consequences following even a short-term loss of income.

The larger part of the sample was concerned about family and close friends, which was associated with an increased risk of symptom-defined PTSD. Having someone close who might be at risk of becoming severely ill or die is considered a significant threat. In the coronavirus context, these concerns may often be related to persons of higher age, such as parents and grandparents, and especially if they have an underlying disease.

The finding that experiencing quarantine or isolation was associated with post-traumatic stress is in line with other studies conducted in the context of previous epidemics [31,32]. Quarantine brings thoughts about possible consequences of the pandemic and the perceived centrality of the event closer to consciousness, which may result in an increase in post-traumatic stress [33,34]. The consequences of social withdrawal or isolation also limit the opportunities for social support, which is essential in counteracting post-traumatic stress reactions [35,36].

The risk of PTSD of those who had been infected with COVID-19 was not significantly different from that of their non-infected counterparts. The finding indicates that getting the infection did not result in much more post-traumatic reactions than the fear of becoming infected in general. However, the study included few cases of infection and it is unknown to what extent the infected have been affected by the disease.

Our cross-sectional study did not provide information about the duration or trajectory of post-traumatic stress in the population. These will probably depend on how the infection spreads in the population, the capacity of the health service, confidence in the authorities’ handling of the epidemic, and whether they will succeed in gaining control. Based on the slow remission of post-traumatic stress, it can be expected that stress reactions will persist in the population for a long time, even with rapid and effective control of the epidemic [9,37]. In the case of new waves of infection, stress reactions can build up to new heights and have an even greater impact on the mental health of the population.

### Study Limitations

We have classified the level of post-traumatic stress as symptom-defined PTSD or not, solely on the basis of the symptom burden reported by each respondent. We have made no attempt to assess the A-criterion in DSM-5, i.e., whether the individual was exposed to threat or danger of such a nature or extent that it justifies a PTSD diagnosis. According to DSM-5 [10], to be diagnosed with PTSD, the person must have been exposed to death or threatened death through direct exposure, as a witness, or have learned that a relative or close friend was exposed to trauma. Despite some objections to the generality of the threat, we have chosen to follow the same line as a number of authors who have estimated PTSD related to the pandemic, whether it is about infected patients [12], health professionals [16], or the general population [28]. The specification that we have measured symptom-defined PTSD is crucial in this regard.

The use of the PCL-5 self-reported inventory, rather than using clinical interviews to assess symptoms of PTSD, also deserves a warning about the interpretation of the results. Compared with the structural clinical interview of DSM-IV, a Norwegian version of PCL is almost equivalent in its ability to estimate the prevalence of PTSD in epidemiological research [38]. Nevertheless, the lack of clinical interviews in the present study involves a limitation with unknown direction of the possible bias.

The study did not consider comorbid psychiatric disorders. As depression and anxiety partly overlap with PTSD symptoms (e.g., problems with sleep and concentration), we cannot fully differentiate between PTSD symptoms and symptoms potentially better accounted for by other mental disorders.

The link to the survey was distributed openly on the internet via social media channels. While this is an effective recruitment strategy, the self-selected sample was not representative of the general population in terms of age, gender, and education. Responders were largely young, well educated, urban, and female, while older, less educated, rural, and male participants were in the minority. Moreover, we cannot exclude the possibility of sampling bias concerned with how the participants felt about the pandemic situation. For example, our findings could be biased if people who were more affected by the COVID-19 situation were more willing to complete the survey [39].

## 5. Conclusions

We found that 19.5% of women and 12.5% of men reported a symptomatic burden consistent with PTSD. Associations with concern for others and for one’s personal economy indicate that reasons for post-traumatic stress in the pandemic context needs to be understood in a wider context than just as a threat to one’s own life and health. This study has shown that the wider context includes the safety and well-being of family and close ones, as well as the economic foundation for one’s life. The study concludes that the traumatizing potential of COVID-19 extends to the whole of society and not only to people who are infected or who have close contact with patients infected with COVID-19.

## Figures and Tables

**Table 1 ijerph-17-09210-t001:** Sample characteristics (*n* = 4527).

Variables	All	Men (*n* = 659)	Women (*n* = 3850)	*p*
	*n* (%)	*n* (%)	*n* (%)	
Age group				<0.001
18–29	1156 (25.5)	96 (14.6)	1058 (27.5)	
30–39	1220 (26.9)	145 (22.0)	1071 (27.8)	
40–49	931 (20.6)	161 (24.4)	765 (19.9)	
50–59	766 (16.9)	147 (22.3)	614 (15.9)	
60–69	354 (7.8)	77 (11.7)	275 (7.1)	
70+	100 (2.2)	33 (5.0)	67 (1.7)	
Education				<0.01
Higher education	3417 (75.5)	464 (70.7)	2942 (76.5)	
High school or lower education	1105 (24.4)	192 (29.3)	906 (23.5)	
Employment status				0.07
Employed/in education/paid leave	3667 (81.0)	517 (78.5)	3137 (81.5)	
Not employed/in education/paid leave	860 (19.0)	142 (21.5)	713 (18.5)	
Cohabitation				<0.01
Lives with spouse or partner	2714 (60.0)	426 (64.6)	2275 (59.1)	
No spouse or partner	1813 (40.0)	233 (35.4)	1575 (40.9)	
Size of place or residence				0.57
<2000 inhabitants	187 (4.1)	32 (4.9)	154 (4.0)	
2000–19,999 inhabitants	1141 (25.2)	159 (24.1)	978 (25.4)	
20,000–99,999 inhabitants	1091 (24.1)	171 (25.9)	915 (23.8)	
100,000 inhabitants or more	2098 (46.3)	296 (44.9)	1795 (46.6)	
Social support				<0.01
Yes	4097 (90.5)	572 (86.8)	3508 (91.1)	
No	420 (9.3)	86 (13.1)	333 (8.6)	
Problems related to COVID-19				
Economic concerns	985 (21.8)	138 (21.0)	844 (22.0)	0.58
Expecting economic loss	1144 (25.3)	202 (30.7)	938 (24.4)	<0.01
Have been infected	65 (1.4)	11 (1.7)	54 (1.4)	0.59
Have been in quarantine or isolation	1278 (28.2)	164 (24.9)	1106 (28.8)	<0.05
Risk of complications	1061 (23.4)	196 (29.8)	860 (22.4)	<0.001
Concern for family or friends	3796 (83.9)	503 (76.3)	3280 (85.3)	<0.001
Symptom-defined PTSD	832 (18.4)	82 (12.5)	745 (19.5)	<0.001

Note. PTSD is post-traumatic stress disorder. Statistical tests are Chi-square tests. Missing data ranged from 0.00% to 0.64%.

**Table 2 ijerph-17-09210-t002:** Associations with current symptom-defined post-traumatic stress disorder (PTSD) (*n* = 4527).

Independent Variables	Unadjusted			Adjusted		
	OR	95%CI	*p*	OR	95%CI	*p*
Higher age group	0.85	0.80–0.90	<0.001	0.89	0.83–0.95	<0.01
Female gender	1.69	1.32–2.16	<0.001	1.71	1.32–2.21	<0.001
Social support	0.42	0.34–0.52	<0.001	0.43	0.34–0.55	<0.001
Economic concerns	2.95	2.51–3.57	<0.001	2.23	1.81–2.75	<0.001
Expecting economic loss	2.20	1.87–2.60	<0.001	1.39	1.13–1.70	<0.01
Have been infected	0.80	0.40–1.57	0.51	0.69	0.34–1.40	0.30
Have been in quarantine/isolation	1.40	1.19–1.65	<0.001	1.31	1.10–1.55	<0.01
Risk of complications	1.64	1.39–1.93	<0.001	1.77	1.47–2.14	<0.001
Concern for family or close friends	2.09	1.63–2.68	<0.001	1.83	1.42–2.38	<0.001
Adjusted model parameters						
Model fit						*p* < 0.001
Cox Snell R^2^						7.0%
Nagelkerke R^2^						11.3%

Note. PTSD is post-traumatic stress disorder. Dependent variable is current PTSD. Social support is having friends that will help if needed. Economic loss is expecting the COVID-19 situation to cause personal economic loss. Risk of complications is self-reported risk of complications in the case of contracting the coronavirus.

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
