# Peer review of "Post-Traumatic Stress Disorder and Associated Factors during the Early Stage of the COVID-19 Pandemic in Norway"

_ijerph, 2020, doi:10.3390/ijerph17249210_

Round 1

Reviewer 1 Report

Review - Posttraumatic stress disorder and associated factors during the early stage of the COVID-19 pandemic in Norway

The aim of the study was to examine post-traumatic stress reactions related to the COVID-19 pandemic,the prevalence of symptom-defined post-traumatic stress disorder (PTSD), and factors associated with post-traumatic stress in the Norwegian population during the early stages of the COVID-19 outbreak (from abstract).

The study was interesting in that it attempted to look at mental health across an entire country population.

Although the sample was fairly large (n=4527),  it may not be representative of the entire Norwegian population.  The statement in the abstract “In conclusion, posttraumatic stress reactions were common in the Norwegian population in the early stages of the COVID-19 outbreak” Goes too far for these results. Should be toned down.

Age groups - Since those in higher age categories were likely more at risk for COVID, should they have been adjusted in analysis? Were they more fearful? And, the sample consisted mostly of young persons (18-39), who were less likely to contract COVID. Authors may want to discuss these limiting factors in discussion section which might affect the data.

Authors may want to consider a sub-analysis on the 65 persons who were infected. PTSD may be higher in this group.

Reviewer 2 Report

Thank you for inviting me to review this interesting study. In fact, it is refreshing to read a Covid19-related study that seems well-prepared and is able to formulate a meaningful conclusion. Strengths of the study are its focus on the general population and the inclusion of a range of variables. The statistical methods seem appropriate. Please find a few comments/questions below. 

The introduction refers to the risk of PTSD or PTSD-symptoms in Covid19 patients, frontline staff, and general population. However, why is it important to know the frequency and characteristics of PTSD in the general population exposed to Covid19? Please articulate the rationale.

PTSD has been studied in the context of single events (such as natural disasters) or prolonged events (such as wars). Probably, Covid19 can be considered a prolonged event. How important is this for the occurrence of PTSD symptoms? Your thoughts?

Page 2, line 85: Rephrase, for example:

The research project, including a link to the survey, was also presented in national and local newspapers.

Page 3, line 93. Please add a sentence to explain what ‘NORPOP’ is. (I thought it was a Norwegian Pop music festival, but I might be wrong).

Line 97: If you asked about gender (or sex?), was this limited to male/female? Or had participants other options? If so, which ones, and how did you manage that in the analysis?

Line 101: The question regarding social support was limited to friends. Often, family are a major source of social support. What was the reason for excluding family from this question?

Line 126: Could you add the Cronbach’s alpha for this sample.

Line 130: Authors explained how they dealt with participants with missing data who could not reach the threshold of PTSD symptoms. Could you also explain what you did with participants with missing data who could reach the PTSD symptoms’ thresholds?

Line 174: the description of social support is different from the one given on line 101.

Lines 181-187: Are there any references to support these statements?

Lines 188-193: Is it also possible that younger people may have more fears for economic hardship, or more concerns for family, compared to older people? I suppose that you can check such hypothetical explanations in your data.

Line 246: While authors have acknowledged the study limitations, it could be formulated more affirmative. For example, the sentence: “it renders researchers unable to fully assess the representativity of the sample” sounds fuzzy as it is very clear that the sample was not nationally representative and biased towards young, female, urban, well-educated participants.

Please check references 7 (left margin) and 12 (highlighting).

I hope that these few comments may help to improve and finalize the manuscript. I wish the authors good luck with the revision and publication of the study.

Reviewer 3 Report

This study examines the prevalence of PTSD in Norway post-covid and the factors predicting PTSD symptoms in the sample. The authors presented clear ideas. They also offered a concise literature review, including recent studies on COVID-19 and PTSD. Here are some suggestions to improve the manuscript:

  • The authors studied some possible factors that may associate with PTSD symptoms post-covid; however, no rationale was given on why they chose these factors. At least a paragraph needs to be added in the introduction to discuss why they suspected that the factors they examined (gender, age, etc.) may be related to PTSD symptoms relating to COVID. Ideally, their discussion should be supported by theories or literature reviews.
  • The first paragraph of the Results section (i.e. the "respondents" or sample characteristics) should be discussed in the method/sample section, not in the results. The results section is dedicated to discussing the findings of their research questions. 
  • Detailed analytic procedure is missing in the manuscript. The authors need to describe the exact test they did (logistic regressions?). Their analytic procedures should be so clear that others can reproduce their findings with their data.  
  • Also, when reporting the findings, they only report OR and the p-value when they should also report any sensitivity analysis, if any, they did to compare subgroups (e.g., did they test whether the difference between age group 18-29 and 30-39 are signifciant?)
  • The results description also needs to be clearer. For example, the authors said that "adjusting for all covariates..." - what are the covariates? They were not mentioned before. Or do they simply mean that they ran the whole logistic model with all the predictors all at the same time? Covariates are usually referred to as variables that are unrelated to the main research questions but "co-vary" with the main variables. 
  • Another example of an unclear expression is when they describe the significant findings. They said that "several problems related to COVID-19 were found to be associated with [a] higher risk of PTSD, such as economic concerns (OR: 2.23, p < 0.001).." Do they mean that higher or lower economic concerns were related to the risk? Technically, people can tell from the OR, but what if someone who is reading it but does not understand OR? Also, when they said "The risk of PTSD was higher for those of younger age" did they treat the age variable as continuous? From Table 1, it seems that they are treating the age variable as categorical/dummy coded, but then in Table 2 they seem to be treating it as continuous?
  • Table 2 should precede the discussion section since it pertains to the results section. 
